# Oxidative Stress and Nutraceuticals in the Modulation of the Immune Function: Current Knowledge in Animals of Veterinary Interest

**DOI:** 10.3390/antiox8010028

**Published:** 2019-01-18

**Authors:** Monica Colitti, Bruno Stefanon, Gianfranco Gabai, Maria Elena Gelain, Federico Bonsembiante

**Affiliations:** 1Department of Agrifood, Environmental and Animal Science, University of Udine, Via delle Scienze 208, 33100 Udine, Italy; monica.colitti@uniud.it (M.C.); bruno.stefanon@uniud.it (B.S.); 2Department of Comparative Biomedicine and Food Science, University of Padova, Viale dell’Università 15, 35020 Legnaro (PD), Italy; gianfranco.gabai@unipd.it (G.G.); mariaelena.gelain@unipd.it (M.E.G.)

**Keywords:** veterinary medicine, reactive oxygen species/reactive nitrogen species (ROS/RNS), immune system, regulation, nutraceutical

## Abstract

In the veterinary sector, many papers deal with the relationships between inflammation and oxidative stress. However, few studies investigate the mechanisms of action of oxidised molecules in the regulation of immune cells. Thus, authors often assume that these events, sometime leading to oxidative stress, are conserved among species. The aim of this review is to draw the state-of-the-art of the current knowledge about the role of oxidised molecules and dietary antioxidant compounds in the regulation of the immune cell functions and suggest some perspectives for future investigations in animals of veterinary interest.

## 1. Inflammation, Leukocyte Recruitment, and Activation

The inflammatory process is the physiological response of the immune system to the stimulation by injuries of various origin. The aims of the inflammatory process are to restore homeostasis and to maintain a constant internal milieu. The complex pathways of inflammatory response are aimed at blocking the agent causing tissue injury, to minimize its effect, and to promote wound healing [1]. The majority of these pathways are highly conserved among domestic mammals [2].

The signals that induce the inflammatory response can be exogenous, like those of microbial origin, or endogenous. The microbial triggers can be the pathogen-associated molecular patterns (PAMPs), a defined set of conserved molecules (e.g., lipopolysaccharide, LPS). PAMPs are shared by different strains of microorganisms, and they are recognized by the host organism through pattern recognition receptors (PRRs), such as the Toll-like receptors (TLRs). Another class of microbial inducers are virulence factors restricted to a defined pathogen family (like exotoxin in Gram positive bacteria), which are not recognized by defined host receptors. These factors induce tissue damages and, thus, trigger the inflammatory response [3]. 

The damage-associated molecular patterns (DAMPs) are compounds derived from damaged or death cells following injuries or non-septic inflammation, which are able to induce inflammation by activating this sensing virulence machinery in a non-specific way [2,3,4]. The endogenous inducers of inflammation are signals produced when a tissue or an organ is somehow stressed, damaged, or malfunctioning. The common pathways by which these factors trigger an acute inflammatory response are mainly related to the loss of the compartmentalization maintained in normal tissues and cells. In turn, this induces the release of molecules, the interaction between epithelial and mesenchymal cells caused by the disruption of the basement membrane and the damage of vascular endothelium, along with the resulting release of plasma proteins in the interstitial space [3]. 

Other endogenous factors are more related to the induction of a chronic inflammation, such as advanced glycation end products (AGEs) and oxidized lipoproteins (high-density lipoproteins, HDL, and low-density lipoproteins, LDL). AGEs can form cross-links with other proteins, leading to the inactivation of their function, or they can interact with their receptors (receptors of advanced glycation end products, RAGE), stimulating several pathways which, in turn, activate the transcription of pro-inflammatory genes [5]. AGEs are produced during both physiological and pathological conditions, such as ageing or diabetes, and in both well and poorly compensated diabetic dogs, plasma AGE concentrations are higher compared to healthy dogs, even if they do not correlate with the clinical improvement [6]. In general, pro-oxidative conditions and the presence of reactive oxygen species (ROS) accelerate AGE accumulation [7]. In the same way, ROS induced the formation of oxidized lipoproteins, which promote pro-inflammatory signals.

After the initial inflammatory trigger, some modifications in the immune system occur in the organism, in order to limit the damage. Typically, the immunity response consists of two distinct, but highly interconnected pathways: the innate and the adaptive response. The innate response is the more conserved reaction during evolution and it aims at giving immediate protection against pathogens or tissue damage [8]. The first step is the recognition of the inflammatory stimuli by the tissue-resident macrophages expressing PRRs, able to recognized PAMPs, oxidized LDL and other endogenous factors [9]. The complex system of cytokines, chemokines, vasoactive amines, and other inflammatory mediators produced by macrophages leads to a massive migration of blood leukocytes, mainly neutrophils, in the site of injury. The increased permeability of blood vessels combined with the interaction between endothelial selectins and leukocytes integrins allows the selective leak of the required blood components [3]. 

Neutrophils represent the first line of defense against the invading pathogens and are the key effectors of the innate immune response. After their activation in tissue, neutrophils perform a number of functions that facilitate pathogen clearance and inflammation resolution. These activities include phagocytosis, generation of ROS, degranulation with release of microbiocidal compounds, and formation of extracellular traps. These mechanisms are powerful, but unspecific, effectors that could severely damage the surrounding tissues [10]. Consequently, this response has to be tightly regulated to avoid detrimental effects.

A more specific response is the adaptive one: its function is to selectively attack the foreign microorganisms, distinguish them from the self-antigens, thus tailoring a reaction against specific antigens avoiding damage to the host cells. This reaction is slower and it is linked to the antigen recognition by specific receptors on lymphocytes. When an antigen is presented to T-lymphocytes, they start the direct response by stimulating B-lymphocytes to produced specific antibodies or by directly attack antigen-bearing cells with cytotoxic T-cells [9]. 

The success of acute inflammation is the elimination of the causative agent and the resolution of the pathological process with the reparation of the tissue damage. The acute inflammation must be promptly controlled to avoid an over-response, which may lead to detrimental consequences. When the anti-inflammatory feedback mechanisms fail to switch-off the inflammatory process, a chronic inflammatory state may occur. Usually, the onset of chronic inflammation is driven by macrophages, both tissues resident and newly recruited. The switch of lipid-derived molecules, from pro-inflammatory prostaglandins to anti-inflammatory lipoxins, helps the tissue-repairing processes by inhibiting the recruitment of neutrophils and promoting the recruitment of monocytes. Monocytes have a crucial role in tissue repair process by removing cellular debris and producing growth factors. In some circumstances, the chronic inflammatory condition is not confined to the originally inflamed tissue, but it assumes the connotation of a chronic challenge for the whole organism. The generation of pro-oxidant chemical species is one of the most evident consequences of inflammation [11].

### 1.1. Inflammation and Oxidative Stress at the Site of Infection

Oxidative stress can affect the immune response at the site of infection/lesion. Neutrophils give a substantial contribution to oxidative stress. As neutrophils possess a high destructive potential against invading pathogens, which can be also directed towards animals’ tissues causing “collateral” damages, their functions should be tightly regulated to avoid unwanted consequences [10,12]. Neutrophils express a number of inhibitory receptors that can negatively regulate their functions. These negative regulatory pathways have been extensively reviewed elsewhere [12] and are beyond the scope of this work. Here, the focus has been placed on the regulation of ROS and reactive nitrogen species (RNS) production, and on the consequences of the inhibitory regulation failure. 

When neutrophils arrive at the inflammatory site, they are exposed to several host and pathogen-derived factors that delay apoptosis and stimulate their functions [12]. In particular, the activation of the enzyme nicotinamide adenine dinucleotide phosphate (NADPH) oxidase (NOX) leads to the generation of a great amount of superoxide, which act as the precursor of hydrogen peroxide and other ROS [13]. Hydrogen peroxide can react with the enzyme myeloperoxidase (MPO), giving rise to a strong oxidant and short-living intermediates capable of reacting with halides (Cl, Br, NO) and generate highly microbiocidal species, such as hypochlorous acid (HOCl). These molecules can be also released to modify extracellular targets and they affect the function of neighboring cells [13]. Thus, large amounts of ROS and RNS act as formidable weapons against invading pathogens. Conversely, low intracellular ROS and RNS serve as intracellular signaling molecules involved in the regulation of several immunomodulatory pathways [13,14].

Neutrophils can exert also a regulatory role on the adaptive immune response and, as reported in a number of studies (reviewed by Christoffersson and Phillipson [10]), neutrophils (or a subset of neutrophils) produce the immunosuppressive cytokine interleukin (IL)-10. Moreover, neutrophils can attenuate the adaptive immunity response through MPO release, which can interact with dendritic cells and inhibit the generation of adaptive immunity response by suppressing dendritic cell activation, antigen uptake/processing by MPO-generated reactive intermediates, to limit T-cell driven tissue inflammation [15].

Under physiological conditions, the equilibrium between ROS and antioxidant systems guarantees the proper functioning of T cells and the mounting of a controlled immune response. However, it is conceivable that altered/excessive extracellular ROS and RNS accumulations can affect the mounting of the immune response and contribute to the generation of a systemic inflammatory status, characterized by oxidative stress (OS). The complex interactions among phagocytic cells, T cells, dendritic cells (DC), and regulatory T cells (Treg) have a key role in the adaptive immune response, and these interactions are profoundly affected by both the intra- and extra-cellular redox environments [16]. 

In neutrophils, the association of the enzyme xanthine oxidase with TLR4 induces the productions of extracellular superoxide that enhances nuclear translocation of nuclear factor κ B (NF κB) and increases expression of NF-κB-dependent pro-inflammatory cytokines [17]. Therefore, the excessive/prolonged release of superoxide is potentially co-responsible of enhancing the systemic inflammation and oxidative stress (Figure 1). In this context, the modality of neutrophil death may be crucial for the evolution of the inflammatory process. In particular, if neutrophil death occurs by cytolysis, this likely implies the amplification of the inflammatory response [13]. It is possible that the accumulation of oxidation products, such as oxidized proteins at an inflammatory site will lead to a progressive reduction of neutrophil viability, as described in cow’s neutrophils [18]. 

Superoxide generated by neutrophils can negatively affect T cell signaling, activation, proliferation and, perhaps, viability. Hydrogen hydroperoxide (H_2_O_2_) can differentially affect the different T cell subsets: memory T cells are the less susceptible and effector T cells the most susceptible to H_2_O_2_ attack. Interestingly, Treg suppressive capacity resists to H_2_O_2_ micro molar levels [16]. 

The extracellular redox potential can affect the intracellular antioxidant systems, which are mainly represented by intracellular thiols (reduced glutathione/ oxidized glutathione, GSH/GSSG, cysteine/cystine, and thioredoxins), and guarantees immediate protection against intracellular ROS/RNS formation, therefore, affecting the intracellular redox balance [14]. An example of these interferences between extracellular and intracellular redox potentials is represented by the cross-talk between DC and T cells described by [19]. Briefly, cysteine is necessary for T cell activation and successive proliferation; however, T cells are depending on DC for cysteine provision. The DC-dependent cysteine release has other consequences including shifting the extracellular ambient redox. The extracellular redox environment influences the equilibrium between oxidized and reduced thiols on the extracellular face of the membrane proteins. Many of these proteins are rich in cysteine and variations of the extracellular redox status may cause functional changes in redox-sensitive proteins, which could be important in the communication between DC and T cells. Interestingly, these authors [19] demonstrated that Treg interfere with this process.

So far, these aspects have not been deeply investigated in domestic animals and should deserve more attention. It is worth noting, however, that differences in the neutrophil biology among animal species are likely to exist, and generalizations about neutrophil functions should be taken with caution. As an example, bovine neutrophils share similarities to those of other species, but they also show functional differences [20]. Therefore, regulatory mechanisms linking inflammation and oxidative stress should be investigated across species. The periparturient dairy cow is one of the most studied models for the association between increased disease susceptibility and oxidative stress. The incidence of metabolic and production diseases, such as milk fever, mastitis, fatty liver disease, ketosis, metritis, and hypomagnesaemia, is higher during the first weeks of lactation. In addition, although evidence indicates that the dairy cow suffers from sub-optimal immune response, knowledge of the dynamics and pathophysiology of immunosuppression encountered during this period is far from being fully understood [21,22]. One potential cause of reduced immune cell functions around parturition can result from the shortage of energy and nutrients, since they are mostly directed to support pregnancy and the onset of lactation [21,23]. 

Inadequate levels of some vitamins and microelements, such as vitamins A, E, and D, and selenium (Se), may negatively affect the immune system of the cow [24,25]. An imbalanced antioxidant level, can affect the interactions between immune and not-immune cells. Endothelial cells (EC) play an important role on neutrophil migration and activation, and an increase in neutrophils’ adherence has the potential to affect inflammation outcome. Interestingly, when neutrophils are moving through tissues to reach the place of infection, NOX is inhibited by an adhesion-mediated mechanism dependent on a variety of extracellular matrix proteins, which delays the ROS formation [12]. Using an in vitro model of Se deficiency in primary bovine mammary artery EC, Maddox and colleagues [26] observed that Se-deficient EC showed significantly enhanced neutrophil adherence when stimulated with tumor necrosis factor α (TNFα), IL-1, or H_2_O_2_. The expressions of the intercellular adhesion molecule 1 (ICAM-1) and E-selectin were enhanced in Se-deficient EC stimulated with TNFα, while P-selectin expression was greater in Se-supplemented EC stimulated with TNFα.

In cattle, clinical and subclinical uterine infections are major causes of infertility. Both immune cells and endometrial cells possess PAMP and DAMP receptors. Once these receptors are activated by PAMP or DAMP, cells produce IL, mainly IL-1, IL-1, IL-6, and IL-8, and prostaglandin E_2_. In turn, these chemo-signals attract neutrophils and macrophages to the site of infection or tissue damage, and the intensity of the inflammatory response should match the severity of the insult [27]. It was hypothesized that possible cause of late embryonic mortality in cows are subclinical uterine infections, which are also responsible of impaired neutrophil function in the peripheral circulation leading to systemic OS. This may explain the increase in plasma advanced oxidized protein product (AOPP), a biomarker of MPO activity, observed in dairy cows affected by late embryonic losses [28,29].

Mammary glands inflammation and the adaptive immune response have been extensively studied in dairy animals. The mammary gland can mount a pathogen-specific immune response, as intramammary infections caused by *Staphylococcus aureus* mostly induce subclinical mastitis, often resulting in a chronic disease, while *Escherichia coli* often induces severe acute clinical mastitis [30]. The typical bacteria involved in mammary infections activate the local immune system in different ways, which can influence the severity of the outcome. *E. coli* infections induce rapid and massive somatic cell count increases, whereas *S. aureus* infections result in a more gradual increase over a period of a few days. Regardless the pathogens involved, the inflammatory response could lead to an increase in OS locally: a preliminary study found a statistically significant correlation between AOPP and the percentage of neutrophils in milk [31]. 

Local inflammatory conditions, in which OS could play an important role, have also been investigated in pet animals. The inflammatory bowel disease (IBD) is characterized by persistent or recurrent activations of the mucosal immune system accompanied by infiltrations of inflammatory cells in the intestinal mucosa [32,33]. It is the most common cause of chronic intestinal disease in dogs, and results in diverse and often debilitating clinical signs [34,35]. Mean concentrations of the antioxidant biomarkers analyzed, with exception of ferric reducing ability of the plasma, were significantly lower and oxidant markers were significantly higher in sera of dogs with IBD than in healthy dogs, even if the source of the systemic ROS has not been identified [36].

### 1.2. Inflammation and Systemic Oxidative Stress

As described in the first section of this review, inflammation is a very complex response to cope with cellular injuries, and it should be resolved as soon as the threat is removed to avoid collateral damage. Oxidative stress is associated to several inflammatory and pathological conditions, which have been the object of numerous reviews [24,37,38,39,40,41,42,43,44]. Recently, the term “oxinflammation” has been proposed to describe the vicious circle linking chronic and systemic oxidative stress to mild chronic inflammation, which may lead to loss of reactivity to mount an adaptive response and predispose to diseases [11]. In companion animals, canine visceral leishmaniasis (CVL) is an example of the systemic effect of OS, which affects the cellular immunity of infected dogs, through impairing lymphoproliferation and microbiocidal mechanisms. Heme oxygenase-1 (HO-1) and its metabolites, OS and IL-10 levels could be related to this impairment. HO-1 is responsible for degrading the hemoglobin’s heme into iron, carbon monoxide, and biliverdin, which is rapidly converted into bilirubin [45]. The compounds derived from heme degradation and IL-10 were elevated in plasma and spleens of infected dogs. IL-10 and HO-1 levels were positively correlated. Inhibition of HO-1 increased lymph node cells’ proliferation and decreased IL-10 and IL-2 production in the presence of Leishmania infantum soluble antigen. The increased HO-1 metabolism observed in CVL is probably associated with OS and increased IL-10, which could be one of the mechanisms responsible for inhibition of the lymphoproliferative response in affected dogs [46].

The levels of OS markers, plasma malondialdehyde (MDA), total antioxidant capacity, whole blood glutathione peroxidase (GPx), and erythrocyte superoxide dismutase (SOD) were also investigated in canine atopic dermatitis. Kapun and colleagues found significantly higher plasma MDA levels in patients than in healthy dogs and a highly positive correlation between the Canine Atopic Dermatitis Extent and Severity Index (CADESI) score and MDA, indicating an association between the severity of canine atopic dermatitis and the extent of oxidative damage to membrane lipids [47].

### 1.3. Adipose Tissue and Oxidative Stress

The excess of adipose tissue accumulation is often associated with a low grade and local chronic inflammation, due to the activation of the innate immune system. The activation of the innate immune system induces a pro-inflammatory status and OS [44], which contributes to the majority of the obesity-associated chronic diseases [48]. The adipose tissue is composed primarily by adipocytes, but contains also other cells types such as fibroblasts, fibroblastic pre-adipocytes, EC, immune cells, and vascular cells in the so-called stromal vascular fraction. These cells are capable of producing hormones and cytokines, called adipokines or adipocytokines, which have endocrine, paracrine, and autocrine actions. The adipokines are responsible of the production of ROS. The ROS activate the immune cells that produce free-radicals (FR), enhancing the inflammatory status [44] and inducing a higher concentration of AOPP [49], a biomarker of MPO and neutrophil activity [38]. 

The most important pro-inflammatory cytokines produced by the adipose tissue are TNFα, IL-1β, and IL-6 [50]. TNFα promotes the systemic acute-phase response via the release of IL-6, the generation of superoxide anions and the reduction in the synthesis of anti-inflammatory cytokines [51], while IL-1β carries out its pro-inflammatory function inducing the production of other pro-inflammatory cytokines, such as IL-6 [52]. IL-6 has a wide range of activities, including the synthesis of other pro-inflammatory cytokines, the transition from acute to chronic inflammatory diseases [53], and the reduction of lipoprotein lipase activity [54]. The pro-inflammatory cytokines produced by adipose tissue stimulate monocytes and macrophages to produce ROS and NOS [50]. The increased amount of ROS enhance the release of pro-inflammatory cytokines [55]. In obese individuals and in animal models of obesity, a large number of macrophages infiltrates adipose tissue. This recruitment is linked to systemic inflammation that, in turn, increases OS in the adipocyte fraction, mitochondrial dysfunction, and endoplasmic reticulum stress leading to insulin resistance [56]. Moreover, intercellular communication between different adipose tissue cell types includes the counter-regulation between adiponectin and TNFα, and between secreted frizzled-related protein 5 (SFRP5) and WNT5a. Under conditions of obesity the pro-inflammatory factors (TNFα and WNT5a) macrophage-derived cells dominate, inhibiting adiponectin, and SFRP5 [57].

The OS in obese patients is furthermore exacerbated by the impairment of the antioxidant mechanisms, such as SOD, GPx, catalase (CAT), vitamin A, E, C, and β-carotene [58]. Free-fatty acids (FFA) play also an important role in the pathogenesis of obese-related diseases. Obese patients have a pathological higher concentration of FFA that impairs glucose metabolism. The altered glucose metabolism induces insulin-resistance [59], favors the accumulation of fat and glucose in liver [60] and muscle [61] and promotes the oxidation of mitochondria and peroxisomes [44]. The damaged mitochondria cannot oxidize fat properly, so there is an increase in the synthesis of triglyceride, in the deposition of ectopic fat [62,63], and in the synthesis of FR and OS with consequent increased damages to mitochondrial DNA, depletion of ATP [64], and lipotoxicity [65]. As a result, the cellular damage increases the production of pro-inflammatory cytokines that elicit the production of ROS in tissues and increases the lipid peroxidation [66].

Additionally, in pet animals, obesity is a common systemic illness with an increasing incidence: according to some authors, in developed countries 40% of dog and 20% of cats have an excessive body mass [67]. As a consequence, diseases related to secretory activity of adipose tissue and the subsequently chronic inflammation are conceivable even though no specific studies in veterinary medicine are present in the scientific literature.

### 1.4. Inflammaging

Aging-related phenomena are conserved among animal species and, in the context of an increasing lifespan of the human population, the comprehension of the key features of the senescence process is of the utmost importance to develop interventions to guarantee a better life quality and health to the aging subjects. This may be true for few animal species, companion animals in particular, which are facing an increase in their lifespan and represent an increasing geriatric animal population requiring veterinary care. Noteworthy, these animals may also represent models to study those aging-related phenomena [68]. In particular, the dog is considered as an elective model for studying aging and immunosenescence, as this species co-evolved with humans, and was exposed to the same environmental challenges. In addition, it is characterized by a wide spectra of phenotypic differences [69].

Ageing is associated with changes in the immune system and to OS [70], a phenomenon called immunosenescence [70]. In humans, immunosenescence is characterized by changes in number, distribution, and/or functions of all the leukocytes’ subpopulations, as well as by altered production of cytokines [71,72]. The total number of monocytes and natural killer cells increases with age [73,74], while the total number of neutrophils and eosinophils are no different between young and old people [75,76]. Changes observed in the immune system of older dogs, cats and horses are virtually identical to those observed in humans [68]. Indeed, the aged dogs display features of immunosenescence similar to those reported in other species [77]. In healthy beagle dogs, the numbers of CD4+ CD45+ and CD8+ CD45RA+ lymphocytes significantly decrease with age [78]. Moreover, aged dogs display a greater neutrophil-to-lymphocyte ratio and lower CD21+ B cells [79].

It has been demonstrated that microbiocidal and chemotactic abilities of neutrophils, superoxide production by eosinophils, the phagocytic function and migration capability of DC and the cytotoxicity and the cytokines and chemokines production of natural killer cells are decreased in aged subjects [74,75,76,80]. Moreover, it has been demonstrated that the monocytes and macrophage functions are decreased in aged mice [81,82].

The production of pro-inflammatory cytokines, such as IL-6, TNFα, IFN-α, and IL-1β, is higher in old, compared to young, people [80,83,84]. All these alterations make old people less respondent to new antigens [85]. The increased release of pro-inflammatory cytokines induces a low-grade chronic inflammatory status, an increase in free radical formation, and consequent OS [86,87]. In the dog peripheral mononuclear cells, the expressions of IL-2, IL-2R, and IL-2 decrease with age, and this may affect the number of naïve T cells [78]. Moreover, in aged dogs CD4+ and CD8+ T lymphocytes produce more cytokines and display a lower proliferative capacity [77]. 

In the aged immune cells, the production of MnSOD, CAT, GPx is decreased [88,89]. OS induces the accumulation of oxidized molecular aggregates that damage proteins, lipids, and carbohydrates and provokes cellular apoptosis [90]. The OS in ageing cells compromise the functionality of subcellular organelles, compartments, and membranes [91]. The changing in the cellular structure is an important factor in the pathogenesis of autoimmunity [90].

Studying the relationships between immunosenescence and OS in the dog can be a difficult task. The dog species displays a remarkable spectra of phenotypic differences [68], and longevity can be very different among canine breeds [92]. Therefore, studies performed in groups of dogs belonging to the same breed, such as the experimental beagle, may show differences in OS biomarkers [93]. Conversely, studies performed in populations of heterogeneous pet dogs may fail to detect differences in OS biomarkers between adult and aged dogs [79,94]. Moreover, the lifestyle of the subjects, in particular the different nutritional habits, can have a substantial impact on the degree of OS [79].

Most research about OS in the aging dog is related to cognitive dysfunctions [79,95]. Indeed, the aged dog shows a number of behavioral problems related to putative cognitive dysfunctions, and accumulation of oxidized macromolecules in the dogs’ brains accounts for one of the causes of cognitive decline [95]. Biomarkers of OS measured in the peripheral blood may not be able to detect this age-dependent increase in OS within the central nervous system [79]. Recently, an impairment of the immune functions of peripheral blood neutrophils and mononuclear cells was observed in mild Alzheimer’s disease human patients in comparison to healthy elderly subjects. This impairment could be mediated by the higher OS in blood cells and isolated neutrophils and the higher release of basal pro-inflammatory cytokines, such as IL-6 and TNF-α [96]. An increase in the count of peripheral monocytes was observed in cognitively-impaired aged dogs, which is difficult to explain [79]. 

A great research interest is given to the use of nutraceuticals and antioxidants to alleviate the cognitive impairment in dogs and, indeed, cognitive performance can be improved by dietary manipulation. However, more research is needed to determine what compounds and/or compound combinations are effective, their dosage and timing of administration, and long-term effects [95].

## 2. Functional Compounds in Animal Nutrition

The term nutraceutical derives from ‘nutrition’ and ‘pharmaceutical’ and many definitions can be found in the dictionaries. On a broader meaning, nutraceutical is a food which provides health benefits other than its nutritional value and that contains bioactive compounds which interact at different levels with the animal physiology [97,98,99,100,101,102,103,104,105,106,107,108,109,110,111,112,113,114]. Probiotic, prebiotic, secondary plant metabolites, amino acids, peptides, fatty acids, and essential oils are a non-exclusive list of nutraceutical compounds and, in this review, we will focus on secondary plant metabolites, probiotics, and prebiotics.

### 2.1. Anti-Oxidant Compounds

The secondary plant metabolites are a very wide group of phytochemicals initially known for their anti-nutritional characteristics, but more recently studied for their role in modulating the biological response of the animals. A very large number of phytochemicals exist, and some reports estimate around 80,000 compounds [97], but new compounds will be likely isolated and identified. According to Wink [115] the plant secondary metabolites can be grouped in nitrogen-containing and not-nitrogen containing compounds (Table 1). Phytochemicals can be also classified into hydrophilic and hydrophobic, and polyphenols, terpenoids, and carbohydrates have attracted the attention of researchers for their bioactivity in animals. 

Terpenoids are very well known compounds, which include provitamins and vitamins, as carotenoids and tocopherols. Polyphenols represent a large family of phytochemicals, including flavonoids, isoflavonids, lignans, stilbenoids, curcuminoids, and tannins and are very common in plant foods [116,117]. Among the polyphenols, flavanols are popular members of flavonoids and include a large variety of monomers, (+)-catechin or (−)-epicatechin, and oligo and polymers (proanthocyanidins). The positive effects of these latter secondary plant metabolites are mainly attributed to their antioxidant activities, including the control of inflammatory processes that can be directly or indirectly related to the compounds.

Plant nutraceuticals has been considered for their direct antioxidant activities, which depends on their chemical reactivity, in relation to the quenching or scavenging of free radicals produced during cell metabolism. The quenching of free radicals means a reduction of electrophilic species, as peroxyl and hydroperoxide radicals (ROO• and ROOH, respectively), while the scavenger activity refers to a reaction of the hydroxyl group of the phenol ring with a reactive oxygen species to form a more stable phenoxyl radical (PhO•) product [118]. This implies that the bioactivity of nutraceuticals requires a direct reaction with the oxidized species and thus depends from their localization (intra- or extra-cellular), solubility (hydrophilic/hydrophobic), adsorption, distribution, metabolism, and excretion [118,119].

These nutraceuticals can interact with the organisms at the cellular and molecular levels, through the regulation of gene expression, protein and DNA repair, and epigenetic controls [120]. Several studies have investigated the molecular and cellular role of nutraceuticals in animal organisms, taking advantage of high-throughput screening. Nutrigenomic studies in humans and laboratory animals have clearly shown that nutraceuticals can modulate gene expression and signaling processes and, among the others cell apoptosis, drug metabolism, immune modulation, and metabolism [121,122,123,124].

Grape polyphenols are considered as potent antioxidants and several studies in human have been conducted, as it can be seen from recently published reviews [125,126]. Even if grape polyphenols can be found in commercial supplements and complete diets for companion animals, in literature no scientific publication in this field are found, while few published studies are available for ruminants [99,100,105,109,110,114]. Interestingly, a common outcome of these studies was a significant increase of antioxidant activity at a blood level, measured as either biochemical markers or gene expressions. In particular, the effect of grape polyphenols resulted in an enhancement of SOD gene expression [99,100,110,127].

Dietary polyphenols and their metabolites exert a beneficial effect through a combination of mechanisms which may include the reduction of inflammation and oxidative stress [128,129], as well as the inhibition of intestinal glucosidases and glucose transporters that reduce postprandial glycaemia [130].

### 2.2. Anti-Inflammatory Compounds

The anti-inflammatory activity of nutraceuticals can modulate the functions of immune cells or the activity at a tissue and organ level, which are involved in the secretion of signaling molecules or ROS and RNS. This modulation was demonstrated in cultured phorbol myristate acetate (PMA) activated ovine neutrophils treated with extracts of *Vitis vinifera*, *Thymus vulgaris*, *Echinacea angustifolia*, *Larix decidua*, and *Andrographis paniculata* [102,131]. The cellular adhesion and SOD production of neutrophils after exposure with nutraceuticals were reduced, supporting a strong anti-inflammatory activity.

Proinflammatory cytokines, other than growth factor and environmental stimuli, activate the NFκB pathway. Among the proinflammatory cytokines, TNFα and IL-1b are known potent activators of NFκB, which enhances the NOX activity in mitochondria, increasing the production of free radicals [132]. In a dietary intervention study in dogs, the effect of the administration of *Vaccinium myrtiullus* and *Curcuma longa* was investigated [113]. After 60 days of administration, these extracts down-regulated the mRNA expressions of TNFα, NFκB and prostaglandin-endoperoxide synthase 2 (PTGS2, also known as COX2) in peripheral blood cells and reduced the concentration of ceruloplasmin.

A proper functioning of the cell involves the interaction between the NF-E2-related factor 2 (Nrf2) and NFκB pathways to resolve inflammatory response and the imbalance between them was related to degenerative diseases and cancer [133]. The Nrf2-mediated signaling plays a pivotal role to coordinate redox balance within the cell, through the regulation of antioxidant and phase II detoxification enzymes, like NADPH, NAD(P)H quinone oxidoreductase 1, GPx, HO-1, and antioxidant genes, as SOD (Figure 2). The involvement of Nfr2 in the anti-inflammatory activity depends upon the Nrf2/HO-1 axis, by the inhibition of the nuclear translocation of NFκB [134], and down-regulation of proinflammatory cytokines [135]. Nrf2 autoregulates its own expression through an antioxidant response elements-like element located in the proximal region of its promoter leading to protracted induction of phase 2 genes in response to chemo-preventive agents [136]. In normal conditions, the transcription activity of Nrf2 is inhibited by ubiquitination and proteasomal degradation via the Kelch domain of the Kelch-like ECH-associated protein 1 (KEAP1) homodimer (Figure 2). After exposure of oxidative stress or to antioxidant compounds, the thiol modification of cysteine residues of KEAP1 release Nfr2 in the cytosol, preventing its degradation [137]. This is not the only mechanism, and KEAP1-independent regulation, and several miRNA, nuclear receptors, such as peroxisome proliferator-activated receptor-γ and glucocorticoid receptors, were reported as inhibitors of Nrf2 [135].

Phytochemicals, as sulforaphane from cruciferous plants, up-regulate Nrf2 in primary peritoneal macrophages, reducing PTGS2, TNFα, and IL-1b after exposure to LPS [138]. The ability of nutraceuticals to interact with the cross-talk bewteen Nrf2 and NFκB pathways has been reported in H_2_O_2_ activated macrophages [108]. In this study, the treatment with pure extracts of *Arctium lappa*, *Camellia sinensis*, *Panax ginseng*, or *Vaccinium myrtillus* enhanced the relative gene expression of Nrf2 and reduced that of NFκB1 and NFκB2. The efficacy of garlic extracts to control Nrf2 and Nrf2-regulated phase II enzymes was also recently reported in dogs by Yamato and colleagues [139], confirming the sensing of Nrf2 to redox conditions of the milieu. 

### 2.3. Anti-Inflammatory Compounds in Obesity

As previously reported, obesity has been associated with increased oxidative stress and inflammation, which contribute to many adverse health consequences. Bioactive components available in plant extracts may have direct effects on adipose tissue, such as stimulating apoptosis to reduce adipocyte number and inhibiting differentiation or modifying gene expression [98]. Furthermore, after the demonstration of functional brown adipose tissue in human adults [140], attention was also being addressed to develop therapies based on the conversion of fat-accumulating white adipose tissue (WAT) into energy-dissipating brown adipose tissue (BAT) [106].

The cellular redox status is important during adipogenesis, thus, the anti-oxidant actions of phytochemicals may contribute to the inhibition of adipocyte cells differentiation [141].

Among the bioactive compounds that can affect adipogenesis, lipolysis or apoptosis, *Rhodiola rosea* (*Rr*) extract and tyrosol, one of the major and well-known phenolic compounds present in *Rhodiola rosea* species, exert antiadipogenic effects, which could reduce OS through a mechanism that involves antioxidant enzyme responses and the pentose phosphate pathway. *Rhodiola rosea* extract and tyrosol inhibited adipogenesis and reduced ROS levels through an increase of superoxide dismutase activity [142]. In an in vitro model, visceral pre-adipocytes were incubated during differentiation with two different extracts of RR. Results indicated a lipolytic and anti-adipogenetic activity of these extracts, through a modulation of genes in adipocyte function and in inhibition of adipogenesis [143]. 

*Curcuma longa* extract is known to be active on OS, inflammation, and cell death pathways [144]. The effect of 20% curcumin extract was significant in modulating apoptosis on omental pre-adipocytes at 10 and 20 days of differentiation, but not on mature cells [145]. On 3T3-L1 pre-adipocytes, curcumin reduced lipid accumulation and differentiation through the inhibition of fatty acid synthase [146] or down-regulation of transcription factors as peroxisome proliferator-activated receptors (Pparγ), CCAAT-enhancer-binding proteins alfa and beta (C/ebpα, C/ebpβ), and Kruppel Like Factor 5 (Klf5) [147]. 

Again, as reported in Colitti and Stefanon [145], other bioactive compounds, such as docosahexaenoic acid (DHA), resveratrol, and caffeine, affect the adipogenesis on visceral pre-adipocytes. Interestingly, the strongest apoptotic effect was induced by *Taraxacum officinale* leaf extract, even though DNA microarray analysis showed that *Taraxacum officinale* extracts added in 3T3-L1 pre-adipocytes regulated the expression of genes and long noncoding RNAs involved in the control of adipogenesis, in brown fat cell differentiation and in diet induced thermogenesis [148]. However, also small, regulatory noncoding RNA (miRNAs) were claimed to modulate the expression of anti-adipogenic genes in *Rosmariunus officinalis*-treated cells. In particular, in primary omental pre-adipocytes let-7f-1, miR-17, miR-503, and miR-30a modulated genes involved in the cell cycle [149].

Against obesity, natural compounds have also been suggested as treatments to promote the conversion of fat-accumulating WAT into energy-dissipating brite (brown in white) cells [114]. The dissipation of energy deriving from the catabolism of fatty acids in form of heat, namely thermogenesis, is driven by mitochondrial proton pump uncoupling protein 1 (UCP1) in BAT. A list of nutritional factors involved in BAT development and recruitment, WAT browning and UCP1 upregulation was extensively reviewed in [106]. Among these, capsaicin, oleuropein, p-synephrine from *Citrus aurantium*, DHA, and conjugated linoleic acid have been studied in in vivo and/or in vitro approaches. The efficacy of these molecules in inducing the plasticity of WAT towards a brown conversion was evaluated in culture cells through different methodological approaches, such as changes in lipid droplets (LDs) morphology [150], protein and gene markers, and mitochondrial dynamics [151]. 

## 3. Conclusions

Different physiological and pathological condition could be the causes of acute and chronic inflammation, such as ageing, metabolic disorders, and infectious diseases. One of the main consequences of inflammation is the induction of oxidative stress, the production of ROS, RNS, AGE, and several other compounds that lead to tissue damage. The complex pathways beyond these processes are widely studied in human medicine and they are assumed to be conserved among mammalian species. However, a more precise definition of species-specific pathway is needed. Equally, the positive effect of nutraceutical compounds and their antioxidant and anti-inflammatory activities in domestic animals have been investigated, but more information on their exact dosage, timing, and real effects on inflammatory status in animals are far from being deeply understood. 

## Figures and Tables

**Figure 1 antioxidants-08-00028-f001:**
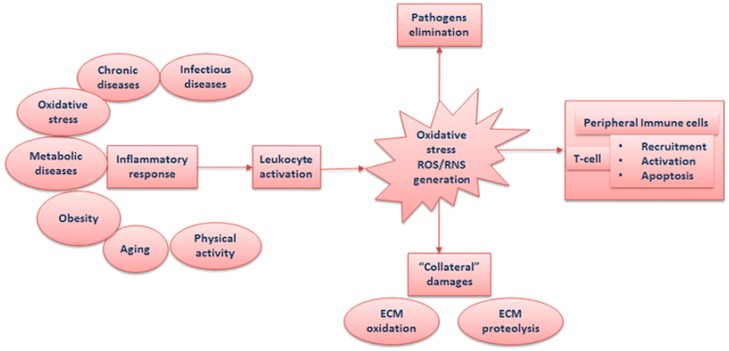
Schematic representation of the relationship among inflammation, oxidative stress, and leukocytes. ROS = reactive oxygen species; ECM = extracellular matrix; RNS: reactive nitrogen species.

**Figure 2 antioxidants-08-00028-f002:**
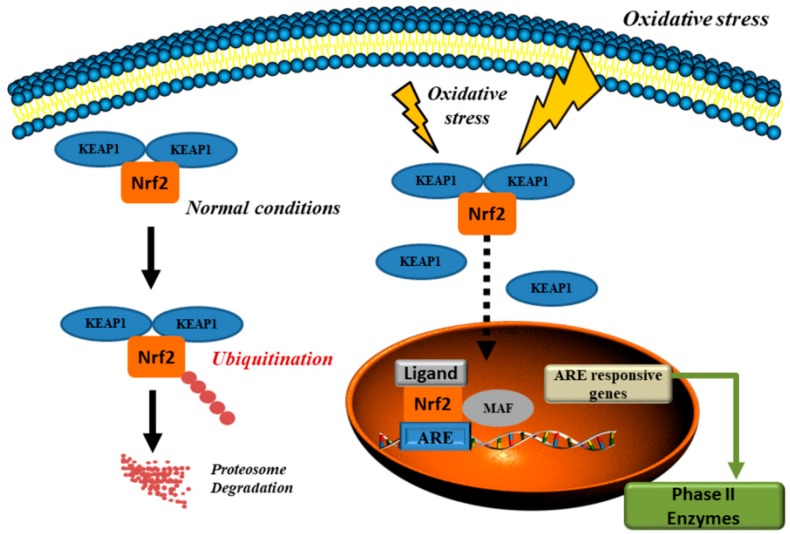
Ubiquitination and proteasomal degradation via the KELCH domain of the KEAP1 homodimer is prevailing in basal conditions. After oxidative stress, Nrf2 is released and translocated to the nucleus where it activates genes with ARE in the promoter region (modified from [135]). KEAP1: Kelch-like ECH associated protein; Nrf2: NF-E2 p45-related factor 2; ARE: antioxidant response element; MAF: musculoaponeurotic fibrosarcoma.

**Table 1 antioxidants-08-00028-t001:** An estimate of plant secondary metabolites [115].

Nitrogen Containing Compounds	Number of Natural Compounds	Non-Nitrogen Containing Compounds	Number of Natural Compounds
Alkaloids	12,000	Monoterpenes	1000
Non protein amino acids	600	Sesquiterpenes	3000
Amines	100	Diterpenes	2000
Cyanogenic glycosides	100	Triterpenes, Saponins, Steroids	4000
Glucosinolates	100	Tetraterpens	350
		Flavonoids	2000
		Polyacetylenes	1000
		Polyketides	750
		Phenylpropanes	1000

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
