# Peer review of "Oxidative Stress and Nutraceuticals in the Modulation of the Immune Function: Current Knowledge in Animals of Veterinary Interest"

_antioxidants, 2019, doi:10.3390/antiox8010028_

Round 1

Reviewer 1 Report

Oxidative stress and nutraceuticals in the modulation of the immune function: current 2 knowledge in animals of veterinary interest. 3

Monica Colitti,2 Bruno Stefanon2, Gianfranco Gabai,1 Maria Elena Gelain,1 Federico 4 Bonsembiante1*

Significance: Are the results interpreted appropriately? Are they significant?

This is a review, so there are no results.

Are all conclusions justified and supported by the results?

The conclusions are supported by previous studies.

Are hypotheses and speculations carefully identified as such?  N/A

Quality of Presentation: Is the article written in an appropriate way?

The article needs to be proofed and written by someone with a much stronger command of English writing and syntax.  I made several edits below, but there are dozens if not hundreds more that need to be made.

Are the data and analyses presented appropriately?

In Figure 1, the authors could add that an additional cause of an inflammatory response is oxidative stress since they discuss that aspect. 

Another Figure could be added to illustrate the changes in the balance of cytokines, cells, enzymes in domestic animals and humans that occur with inflammaging beginning on page 12

In Figure 2, the two drawings should be switched so that the normal condition is shown to the left and the oxidative stress condition is shown to the right.

Maybe a Table showing the nutraceuticals discussed, source of the nutrient, what benefits they possess, and proposed mechanism?

Are the highest standards for presentation of the results used?

Authors need to include a comprehensive listing of all abbreviations.  In the text, abbreviations should follow, not precede the full spelling.

Scientific Soundness: is the study correctly designed and technically sound? N/A

Are the analyses performed with the highest technical standards? N/A

Are the data robust enough to draw the conclusions? Yes, if data means review of the literature.

Are the methods, tools, software, and reagents described with sufficient details to allow another researcher to reproduce the results? N/A

Interest to the Readers: Are the conclusions interesting for the readership of the Journal?

Yes, however, in some areas, the authors go into too much detail and lose the main point.

Examples are lines 461-473,

Will the paper attract a wide readership, or be of interest only to a limited number of people? (please see the Aims and Scope of the journal)

As a review article, it will hold moderate to broad interest.

Overall Merit: Is there an overall benefit to publishing this work?

If the writing style concerns are addressed by using an English language writer.

Does the work provide an advance towards the current knowledge?

It is a review article, so this is N/A.

Do the authors have addressed an important long-standing question with smart experiments? No.

English Level: Is the English language appropriate and understandable?

As discussed above.

First sentence in Abstract lines 13/14: several published researchers many papers deal with the

relationships between inflammation and oxidative stress.

Lines 26/27: The complex pathways of the inflammatory response is are aimed at blocking the agent causing tissue injury, to minimize its effect and to promote wound healing [1].

Line32/33: (e.g. lipopolysaccharide (LPS),

Line 55: in turn

Line 57: suggests that higher AGEs correlate with clinical improvement.  This is not true.

Line 80: Consequently, this response had has to be tightly regulated to avoid detrimental effects.

Line 82/83: its function is to selectively attack the foreign microorganisms, distinguish them from the self-antigens,

Line 118: strong oxidant and short-living intermediate capable of reacting with halides

Line 190: NOX is inhibited by an adhesion-mediated mechanism dependent by on a variety of extracellular matrix proteins, which delays the ROS formation

Line 236: Although it is still under debate whether oxidative stress is the cause or an epiphenomenon of a specific disease, in both humans and animal species of veterinary interest.  This is not a complete sentence, the first part is a clause, the second part is a phrase.

Line 265: The excess of adipose tissue accumulation is often associated to with a low grade

Line 271: These cells are capable to of producing hormones and cytokines,

Line 325: Ageing is associated to with changes in the immune system and to OS

Author Response

Point 1:The article needs to be proofed and written by someone with a much stronger command of English writing and syntax.  I made several edits below, but there are dozens if not hundreds more that need to be made.

Response 1: We checked the text to improve the standard of english and to eliminate the grammatical errors, but we had not enough time to ask the support of an english language writer.

Point 2: In Figure 1, the authors could add that an additional cause of an inflammatory response is oxidative stress since they discuss that aspect. 

Response 2: Done

Point 3: Another Figure could be added to illustrate the changes in the balance of cytokines, cells, enzymes in domestic animals and humans that occur with inflammaging beginning on page 12.

Response 3: A figure with all these information was too complicate to design and for the interpretation, so we did not add any new figure

Point 4: In Figure 2, the two drawings should be switched so that the normal condition is shown to the left and the oxidative stress condition is shown to the right.

Response 4: Sorry but we did not have enough time to change the figure. If the reviewer think that the switch is fundamental we will send the modified figure in the next days.

Point 5: Maybe a Table showing the nutraceuticals discussed, source of the nutrient, what benefits they possess, and proposed mechanism?

Response 5: We added the table in the text (page 19)

Point 6: Are the highest standards for presentation of the results used?

Authors need to include a comprehensive listing of all abbreviations.  In the text, abbreviations should follow, not precede the full spelling.

Response 6: Done, we added the list of abbreviations after the keywords

Point 7: Overall Merit: Is there an overall benefit to publishing this work?

If the writing style concerns are addressed by using an English language writer.

Response 7: We checked the text to improve the standard of english, but we had not enough time to ask the support of an english language writer. 

Point 8: First sentence in Abstract lines 13/14: several published researchers many papers deal with the

relationships between inflammation and oxidative stress.

Response 8: Done

Point 9: Lines 26/27: The complex pathways of the inflammatory response is are aimed at blocking the agent causing tissue injury, to minimize its effect and to promote wound healing [1].

Response 9: Done

Point 10: Line32/33: (e.g. lipopolysaccharide (LPS),

Response 10: Done

Point 11: Line 55: in turn

Response 11: Done

Point 12: Line 57: suggests that higher AGEs correlate with clinical improvement.  This is not true.

the revisor is correct, we misinterpreted the paper. Sorry for the mistake, now the sentences is correct

Line 80: Consequently, this response had has to be tightly regulated to avoid detrimental effects.

Response 12: Done

Point 13: Line 82/83: its function is to selectively attack the foreign microorganisms, distinguish them from the self-antigens,

Response 13: Done

Point 14: Line 118: strong oxidant and short-living intermediate capable of reacting with halides

Response 14: Done

Point 15: Line 190: NOX is inhibited by an adhesion-mediated mechanism dependent by on a variety of extracellular matrix proteins, which delays the ROS formation

Response 15: Done

Point 16: Line 236: Although it is still under debate whether oxidative stress is the cause or an epiphenomenon of a specific disease, in both humans and animal species of veterinary interest.  This is not a complete sentence, the first part is a clause, the second part is a phrase.

Response 16: We deleted the phrase

Point 17: Line 265: The excess of adipose tissue accumulation is often associated to with a low grade

Response 17: Done

Point 18: Line 271: These cells are capable to of producing hormones and cytokines,

Response 18: Done

Point 19: Line 325: Ageing is associated to with changes in the immune system and to OS

Response 19: Done

Reviewer 2 Report

The work is absolutely interesting because it deals in an original way and from a different point of view from the usual reviews available the role of the oxidizing molecules and the antioxidant components of the diet of animals of veterinary interest in relation to the modulation of the immune system.

All of this is appropriately explained in detail in the abstract and in the manuscript in an appropriate way.

The work is well structured and organized in each section, however I believe that some compounds with antioxidant activity should be treated in more detail in a broad review like this and others not even considered to be included.

Something more should be added about the potent antioxidant activity of Rosmarinus officinalis and other aromatic plants and their essential oils.

I suggest some bibliographical references only as an example:

Rassem, H. H., Nour, A. H., & Yunus, R. M. (2018). Biological activities of essential oils–A review. Pacific International Journal1(2), 01-14.

Keni, S., Nambiar, S., Philip, P., & Shetty, S. (2018). A comparison of the effect of application of sodium ascorbate and amla (Indian gooseberry) extract on the bond strength of brackets bonded to bleached human enamel: An In vitro study. Indian Journal of Dental Research29(5), 663.

Herath, H. M. I. C., Wijayasiriwardene, T. D. C. M. K., & Premakumara, G. A. S. (2018). In vitro antioxidant and in vivo anti-inflammatory activity of Curcuma albiflora Thw. Sri Lankan Journal of Biology3(1).

Moreover, other interesting publications can instead serve to enrich the discussion that could be improved by adding some considerations related to other natural compounds.

Mirza, S., Shah, K., Patel, S., Jain, N., & Rawal, R. (2018). Natural Compounds as Epigenetic Regulators of Human Dendritic Cell-mediated Immune Function. Journal of Immunotherapy41(4), 169-180.

Kottuparambil, S., Thankamony, R. L., & Agusti, S. (2019). Euglena as a potential natural source of value-added metabolites. A review. Algal Research37, 154-159.

I believe that the manuscript can be published after these slight revisions and I congratulate the authors on the theme chosen and the scientific rigor with which they have dealt with the research as well as the drafting of the work.

Author Response

Point 1: Something more should be added about the potent antioxidant activity of Rosmarinus officinalisand other aromatic plants and their essential oils.

Response 1: We changed the text
